# Hydrogen Trapping Behavior in Vanadium Microalloyed TRIP-Assisted Annealed Martensitic Steel

**Xiongfei Yang [1,2], Hao Yu [1,\*], Chenghao Song [1,3] and Lili Li [1]**

[1] School of Materials Science and Engineering, University of Science and Technology Beijing, Beijing 100083, China

[2] State Key Laboratory of Vanadium and Titanium Resources Comprehensive Utilization, Panzhihua 617000, China

[3] School of Mechanical Engineering, Dongguan University of Technology, Dongguan 523808, Guangdong, China

[\*] Correspondence: yuhao@ustb.edu.cn; Tel.: +86-010-6233-6588

**Abstract:** Transformation induced plasticity (TRIP)-assisted annealed martensitic (TAM) steel combines higher tensile strength and elogangtion, and has been increasingly used but appears to bemore prone to hydrogen embrittlement (HE). In this paper, the hydrogen trapping behavior and HE of TRIP-assisted annealed martensitic steels with different vanadium additions had been investigated by means of hydrogen charging and slow strain rate tensile tests (SSRT), microstructral observartion, and thermal desorption mass spectroscope (TDS). Hydrogen charging test results indicates that apparent hydrogen diffusive index $D_a$ is $1.94 \times 10^{-7}$/cm$^2$·s$^{-1}$ for 0.21 wt.% vanadium steel, while the value is $8.05 \times 10^{-7}$/cm$^2$·s$^{-1}$ for V-free steel. SSRT results show that the hydrogen induced ductility loss $I_D$ is 76.2% for 0.21 wt.%V steel, compared with 86.5% for V-free steel. The trapping mechanism of the steel containing different V contents is analyzed by means of TDS and Transmission electron microscope (TEM) observations. It is found out that the steel containing 0.21 wt.%V can create much more traps for hydrogen trapping compared with lower V steel, which is due to vanadium carbide (VC) precipitates acting as traps capturing hydrogen atoms.The relationship between hydrogen diffusion and hydrogentrapping mechanism is discussed in details.

**Keywords:** TRIP-assisted annealed martensitic steel; hydrogen embrittlement; hydrogen trapping; thermal desorption spectroscope

## 1. Introduction

Modern advanced ultra-high strength steel (AHSS) had been widely used in various applications such as automotive, engineering, and machinery. However, hydrogen induced delayed fracture had always present a great challenging to the AHSS due to its inherent characteristic of high sensitivity to the hydrogen invasion and enrichment in metal matrix, when this kind of steel had been cold worked or serviced in a moist atmosphere. Extensive efforts had been done in order to address the problem challenged by hydrogen induced delayed fracture behavior. Extensive works showed that introduction of various kinds of hydrogen traps could effectively improve hydrogen induced delayed fracture behavior occurring in high strength steels [1–7]. Effective hydrogen traps including MnS, dislocation, retained austenite, and microalloying elements precipitates such as NbC, TiC, and NbN have been investigated and classified into reversible or irreversible traps. Retained austenite in transformation induced plasticity (TRIP) steel [8] could play an effective role in improving the ductility of high strength steel due to the so-called TRIP effect. Meanwhile, TiC had been well established as a

strong hydrogen trapping sites. Recent researches on vanadium-bearing steels suggested that low temperature vanadium precipitate, usually in form of $V_4C_3$, could effectively bring the ultra-high strength steel with higher resistance to hydrogen induced delayed fracture [7,9–12].

In this work, hydrogen trapping behavior in TRIP-assisted annealed martensitic steels with different vanadium additions have been investigated by means of hydrogen charging and slow strain rate tensile tests (SSRT), microstructral observartion, and TDS. The reasons for choosing TAM steel in this work are:1)on the one hand, it is highly prone to hydrogen embrittlement due to its martensitic matrix and 2) howevere, on the other hand, it is capable of capturing hydrogen since a certain volume of asutenite presents in the martensite matrix.

TAM steel had been first developed by K. Sugimoto et al [13]. They found out that this kind of steel had an excellent combination of higher strength and good ductility. However, TAM steel was highly prone to hydrogen embrittlement due to its martensitic matrix [14] in which the solubility of hydrogen atom is rather low. There are few reported investigations on hydrogen embrittlement of TAM steel and how to improve its hydrogen induced delayed fracture behavior. Therefore, understanding hydrogen trapping behavior and hydrogen embrittlement properties of TAM steels is important to apply this kind of steel for certain applications such as automotive structural parts.

To obtain the required microstructure, TAM steel has to subject to a complex heat-treatment cycle during which a low temperature temping process is necessary. For three major mircoalloying elements such as Nb, Ti, and V, only V is capable of precipitating at around 400 °C of tempering temperature. The low-temperature precipitated VC particles not only further enhances the strength but improves the resistance to hydrogen embrittlement [9] of TAM steel. However, VC precipitates will consume carbon and consequently volume fraction of retained austenite decreases, which in turn deteriorates the TRIP effect of TAM steel. Thus, appropriate vanadium addition in steel is crucial to achieve the balanced combination of high TRIP effect and excellent resistance to hydrogen embtittlement for a given base chemisty, particular to the carbon content.

## 2. Experimental Procedure

### 2.1. Materials

Previous works [4] indicate that small-sized and dispersed VC particles can effectively act as hydrogen trapping sites. It is well known that vanadium can easily fix with nitrogen and thus form vanadium nitride particles at austenite temperature zone. During annealing process the pro-existing VN particles will further coarse and certainly consume vanadium, thus leaving less vanadium available for low temperature VC precipitates. When considering alloying design for this work, nitrogen will maintain as low as possible.

Four steels with various vanadium additions are used in this study. The base chemistry is 0.2%C–1.50%Si–2.0%Mn with similar level of Titanium addition for all steels. Ti addition is desired to fix free N and then no free N available to consume vanadium at high temperature. And all four heats were melted in a lab-scale vacuum induction furnace, and then casted into 50 kg ingots. The ingots were reheated up to 1200 °C then holding for 1 h and hot-rolled into 4 mm thick plates through several passes with final rolling temperature of 800 °C. Then the plates were cold rolled into 2 mm thick sheets. Chemical compositions of studied steels are given in Table 1.

**Table 1.** Chemical compositions of used steels (wt. %).

| Elements | C | Si | Mn | P | S | V | Ti | N |
|---|---|---|---|---|---|---|---|---|
| TAM-V0 | 0.19 | 1.42 | 2.02 | 0.007 | 0.005 | - | 0.030 | 0.0037 |
| TAM-V5 | 0.20 | 1.53 | 2.10 | 0.006 | 0.005 | 0.052 | 0.024 | 0.0036 |
| TAM-V10 | 0.20 | 1.50 | 2.10 | 0.005 | 0.006 | 0.098 | 0.030 | 0.0039 |
| TAM-V20 | 0.20 | 1.54 | 2.05 | 0.007 | 0.005 | 0.21 | 0.029 | 0.0045 |

Critical points measured by a Format II dilatometer are shown in Table 2, where the calculated martensite starting transformation points are given by the Equation (1). During the dilatometer test, a standard specimen of 3 mm in diameter and 10 mm in height was heated up to 920 °C at a rate of 10 °C/s, and then held for 10 min. For the determination of $Ac_1$ and $Ac_3$ points the heated specimen was slowly cooled down to room temperature at a cooling rate of 1 °C/s.

$$Ms(°C) = 561 - 474 \times C(\text{wt. \%}) - 33 \times Mn(\text{wt. \%}) - 17 \times Ni(\text{wt. \%}) - 17 \times Cr(\text{wt. \%}) - 21 \times Mo(\text{wt. \%}) \quad (1)$$

**Table 2.** Critical points measured by dilatometer and the calculated $Ms_C/°C$.

| Critical Points | TAM-V0 | TAM-V5 | TAM-V10 | TAM-V20 |
|---|---|---|---|---|
| $Ac_1$ | 744 | 740 | 745 | 750 |
| $Ac_3$ | 865 | 870 | 875 | 878 |
| $M_{sC}$ | 402 | 398 | 397 | 398 |

The cold rolled sheet had been subject to the heat-treatment cycles indicated in Figure 1. All the heating rates used in the cycles were 10 °C/s and cooling(quenching) rate was 50 °C/s, respectively. Optical, Scanning Electron Microscope (SEM) and TEM observations demonstrated the microstructures consist of annealed martensitic matrix with 11%–15% volume fraction of retained austenite depending on the vanadium contents for different steels.

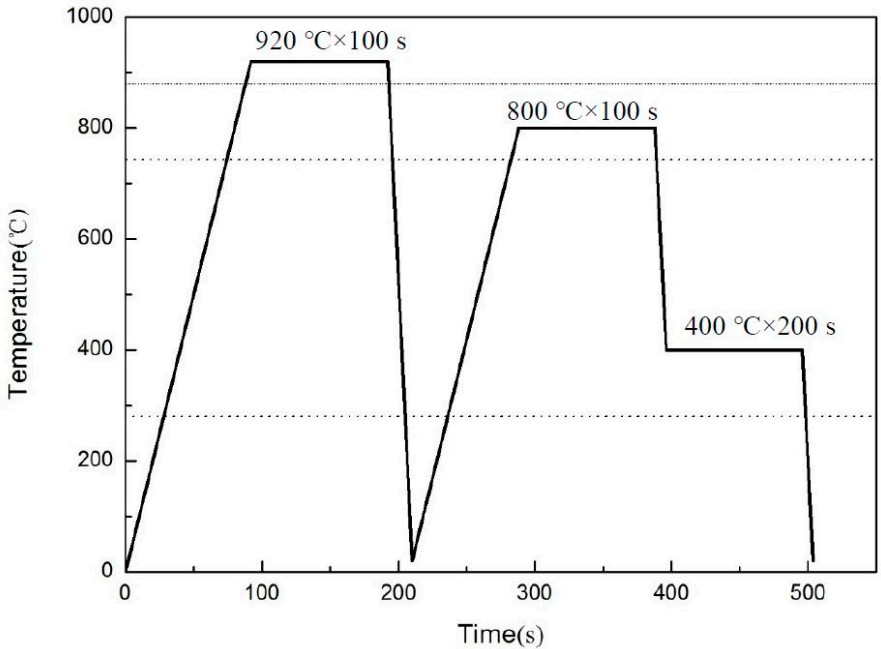

**Figure 1.** Treatment cycles for producing transformation induced plasticity (TRIP)-assisted annealed martensitic (TAM) steels.

## 2.2. Hydrogen Charging Method

Hydrogen charging test was conducted in a standard Devanathan–Stachurski double electrode cells shown in Figure 2, where the thickness of used specimen was 0.3 mm. In order to prevent the surface of specimen anode from dissolution, the anode side of specimen had been chemically deposited by a layer of 200 nm nickel film. Hydrogen was electrically charged in an electrolyte of 0.5 mol/L $H_2SO_4$ solution with addition of 0.22 g/L thiourea ($CH_4N_2S$) at room temperature [15]. The applied cathodic charging current was 5.0 mA/cm². The solution used in the hydrogen detecting cell was

0.2 mol/L NaOH. The hydrogen charging process was immediately stopped when steady state of the permeation rate was achieved, i.e., a constant anodic current was detected.

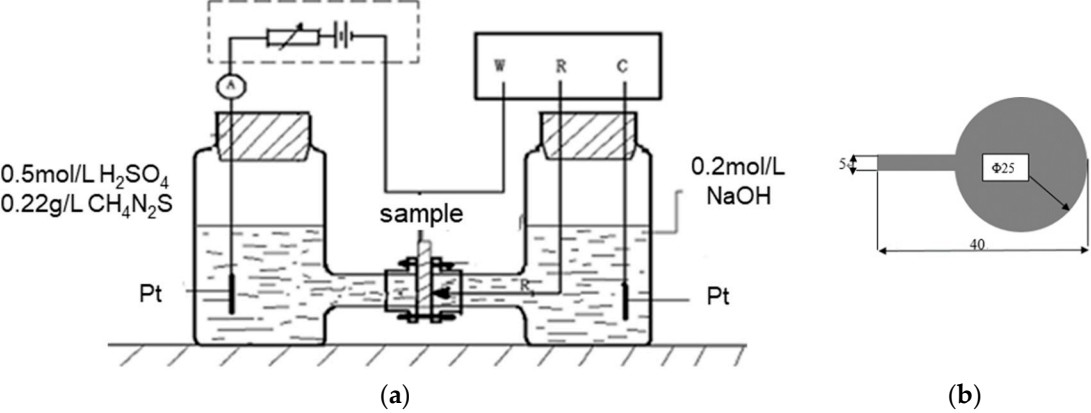

**Figure 2.** (**a**) Devanathan–Stachurskidouble electrode cells used for hydrogen charging test, (**b**) Geometry of tested specimen with 0.3 mm thickness.

### 2.3. Slow Strain Rate Tensile Tests (SSRT)

Slow strain rate tensile tests were conducted on a Chinese manufactured Xi'an LETRY stress-corrosion cracking tester at room temperature. Tensile specimens of 140 mm gauge length and 30 mm gauge width were machined from the TAM treated sheets parallel to the rolling direction. The specimen was immerged into the electrolyte of 0.5 mol/L $H_2SO_4$ solution with addition of 0.25 g/L NaAsO$_3$ [16]. The remained section except gauge was covered by type 704 Silica gel in order to avoid hydrogen entry. During the slow strain rate tensile test the specimen was electrochemically hydrogen charged. A 500 N pre-load was applied to specimen before SSRT to eliminate any possible gap between specimen and tester. The constant applied strain rate was $1 \times 10^{-6}$/s. The cathodic charging current was 5.0 mA/cm$^2$ and maintained until specimen fracture. After SSRT test the specimen was removed from the corrosion product and cleaned with deionized water for the purpose of further thermal desorption analysis.

### 2.4. Hydrogen Analysis

Thermal desorption analysis (TDA) of hydrogen had been conducted for steels with various vanadium additions by a UK manufactured Markes TD100 TDS. The samples were taken from the deformed parts of tensile specimens. During thermal desorption the sampling time was in 5-min intervals and the heating rate was 100 °C/h. The temperature range investigated was from 40 °C to 800 °C. Relatively small specimen cut from the SSRT sample was employed in order to reduce the effect of hydrogen diffusion in the specimen as low as possible.

## 3. Results

### 3.1. Microstructural

The SEM images of TAM steels with various vanadium contents are given in Figure 3. The regular annealed martensites are visible for all four steels. The 0.098%V steel has the finest microstructure compared with other three steels. It has regular martensite plate with width around 100–400 nm. XRD results shows that the studied steels have 11%~15% retained austenite depending on vanadium contents which will be discussed lately.

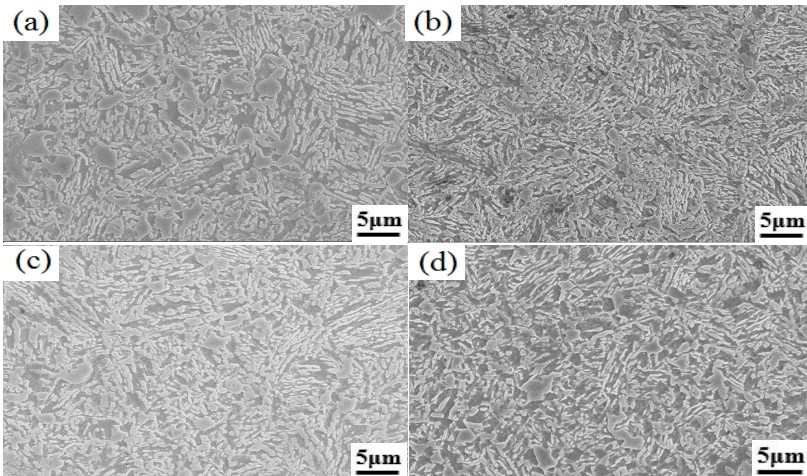

**Figure 3.** SEM images of TAM steels with various vanadium contents. (**a**) 0.052%V, (**b**) 0.098%V, (**c**) 0.21%V, (**d**) 0%V.

### 3.2. Hydrogen Permeation Test

Figure 4 shows the anodic current changing as function of hydrogen charging time for four kinds of steels with different vanadium contents. It is apparent from this figure that at the beginning of the hydrogen charging process, anodic current is at a low level of steady state. After certain amount of charging time, the current increases sharply, which indicates that hydrogen atoms have completely permeated through the anodic side of the tested specimen, but the maximum hydrogen atom flow is not yet achieved. As the charging process continues the current increases and then maintains stability again at a higher level. After this value, the anodic current does not vary with charging time. The second stable current value, referred to as $I_\infty$, indicates steady state hydrogen diffusion achieved in the metal matrix.

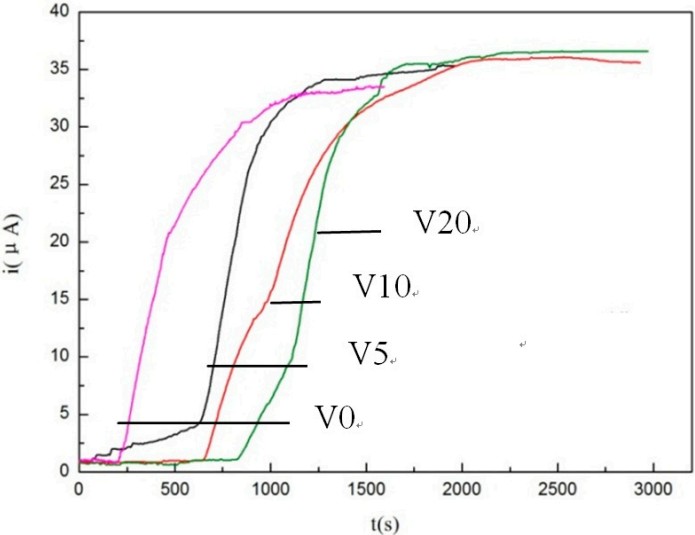

**Figure 4.** Current changing as function of hydrogen charging time for steels with different vanadium contents.

Usually, $t_{0.63}$, corresponding to the time where $I = 0.63I_\infty$ obtained from the permeation curve, stands for the time when the hydrogen atom has completely diffused toward the anodic side of tested specimen. The hydrogen permeation results for different steels are given in Table 3.

**Table 3.** Hydrogen permeation results.

| Steels | Sample Thickness L/mm | $I_\infty$/μA | $0.63I_\infty$/μA | $t_{0.63}$/s |
|---|---|---|---|---|
| TAM-V0 | 0.31 | 33.05 | 20.82 | 199 |
| TAM-V5 | 0.31 | 34.81 | 21.93 | 357 |
| TAM-V10 | 0.32 | 35.85 | 22.59 | 644 |
| TAM-V20 | 0.31 | 36.5 | 23.00 | 826 |

For the vanadium free steel TAM-V0, $t_{0.63}$ is 199 seconds. As the vanadium addition increases, $I_\infty$ slightly increases, but the difference in $t_{0.63}$ is significant (seen in Table 3). In other words, for steel with higher vanadium additions, hydrogen atoms will take a longer time to completely permeate through the specimen.

Now these values shown in Table 3 are used to calculate apparent diffusion coefficient $D_a$, steady-state hydrogen diffusion flow $J_\infty$ and hydrogen concentration in crystal lattice $C_H^S$.

The apparent diffusion coefficient of hydrogen in matrix $D_a$ in unit of $\mathrm{cm^2 \cdot s^{-1}}$ can be given by Equation (2):

$$D_a = \frac{L^2}{6t_{0.63}},\tag{2}$$

where $L$ is specimen thickness in cm.

The steady-state hydrogen diffusion flow $J_\infty$ in unit of $\mathrm{mol/cm^2 \cdot s}$ can be given by Equation (3):

$$J_\infty = \frac{I_\infty}{FA},\tag{3}$$

where $F$ is Faraday constant ($F$ = 96485.3383 A·s/mol), A is specimen area being exposure to electrolyte. In this study, $A$ for all steels is 0.785 $\mathrm{cm^2}$.

The hydrogen concentration in crystal lattice $C_H^S$ in unit of $\mathrm{mol/cm^3}$ is predicated by Equation (4):

$$C_H^S = \frac{J_\infty L}{D_a}.\tag{4}$$

The calculated hydrogen diffusion parameters for different steels are shown in Table 4. It can be seen from the Table 4 that vanadium-free steel TAM-V0 has a higher $D_a$ value while $C_H^S$ value is lower compared with other three vanadium-added steels. For vanadium microalloyed steels, the $D_a$ decreases while $C_H^S$ increases with increasing vanadium content. It is suggested that vanadium addition in steel retards the hydrogen atom diffusion in the matrix.

**Table 4.** Calculated hydrogen permeation parameters.

| Steels | $D_a \times 10^{-7}/\mathrm{cm^2 \cdot s^{-1}}$ | $J_\infty \times 10^{-10}/\mathrm{mol \cdot cm^{-2} \cdot s^{-1}}$ | $C_H^S \times 10^{-4}/\mathrm{mol \cdot cm^{-3}}$ |
|---|---|---|---|
| TAM-V0 | 8.05 | 4.36 | 1.68 |
| TAM-V5 | 4.49 | 4.59 | 3.18 |
| TAM-V10 | 2.65 | 4.73 | 5.72 |
| TAM-V20 | 1.94 | 4.82 | 7.70 |

Besides, from Table 4 it seems look like all four steels present identical $J_\infty$ level. It shall be highlighted that $J_\infty$ stands for steady-state hydrogen diffusion flow, i.e., all available hydrogen traps in the matrix has been 'occupied' by hydrogen atom and then hydrogen atoms freely move through the specimen under the charging current. Identical $J_\infty$ values suggest that these four studied steels consist of identical microstructure.

## 3.3. SSRT Behaviors

The slow strain rate tensile test results for steels under both air media (without hydrogen charged) and electrolyte (with hydrogen charged) are shown in Table 5. The calculated hydrogen induced tensile strength loss ratio $I_T$ and ductility loss ratio $I_D$ based on the measured SSRT tensile properties are also given in this table. Hydrogen induced tensile strength loss ratio $I_T$ is given by following equation:

$$I_T = (R_m{}^o - R_m)/R_m{}^o \times 100\%, \tag{5}$$

where $R_m{}^o$ is SSRT tensile strength for specimen without hydrogen charging, and $R_m$ is SSRT tensile strength for specimen of same steel with hydrogen charging.

**Table 5.** Slow strain rate tensile tests (SSRT) tensile results.

| Steels | Charging Conditions | $R_m$/MPa | A/% | $I_T$ | $I_D$ |
|--------|--------------------|-----------|------|-------|-------|
| TAM-V0 | Uncharging | 1002 | 28.93 | 37.4 | 86.5 |
|        | 5 mA/cm$^2$ | 760 | 3.87 | | |
| TAM-V5 | Uncharging | 1106 | 25.70 | 29.8 | 83.4 |
|        | 5 mA/cm$^2$ | 776 | 4.26 | | |
| TAM-V10 | Uncharging | 1224 | 21.33 | 30.6 | 76.2 |
|         | 5 mA/cm$^2$ | 850 | 5.08 | | |
| TAM-V20 | Uncharging | 1191 | 18.67 | 31.2 | 76.2 |
|         | 5 mA/cm$^2$ | 819 | 4.45 | | |

Hydrogen induced ductility loss ratio $I_D$ is given by Equation (6).

$$I_D = (A^o - A)/A^o \times 100\%, \tag{6}$$

where $A^o$ is SSRT total elongation for specimen without hydrogen charging, and $A$ is SSRT total elongation for specimen of same steel with hydrogen charging.

For all steels, both the tensile strength and total elongation significantly decrease while samples are subject to electrochemically hydrogen charging during tensile test. Vanadium free steel TAM-V0 shows the highest ductility loss, where the absolute reduction value ($A^o - A$) is 25.06% and $I_D$ is 86.5%. As vanadium addition increases, the hydrogen induced ductility loss decreases, regardless of absolute reduction value and $I_D$. However, it is interesting to point out that when vanadium content is increased from 0.098% to 0.21%, the resistance to ductility loss ratio cannot be improved any longer, though the absolute reduction value decreases from 16.25% to 14.22%. This probably originates from the lowest volume fraction of retained austenite in 0.21%V steel, which will be discussed in the later.

## 3.4. TDA Results

The thermal hydrogen desorption rate over time for three vanadium-added steels are plotted in Figure 5a. Peak temperature of hydrogen desorption rate for 0.052%V, 0.098%V, and 0.21%V steels are 167 °C, 169 °C, and 174 °C, respectively. Since no second peak temperature of up to 800 °C is observed for all three steels, the temperature range indicating in the horizontal axis of Figure 5a is then shortened to the 400 °C range in order to clearly distinguish the three curves for different steels. It is clear from the Figure 5a that there is no apparent difference in peak temperature for three vanadium microalloyed steels. However, TDA results show that the amount of desorption hydrogen during heating process are 7 ppm, 8.6 ppm, and 11.25 ppm, respectively, for the above three vanadium-added steels. This implies that desorption hydrogen contents present somewhat increasing tendency as vanadium addition increases in steels. The desorption hydrogen content (assumed as total trapped hydrogen content, $H_{total}$) as function of vanadium content is plotted in Figure 5b, which shows a clear linear relation.

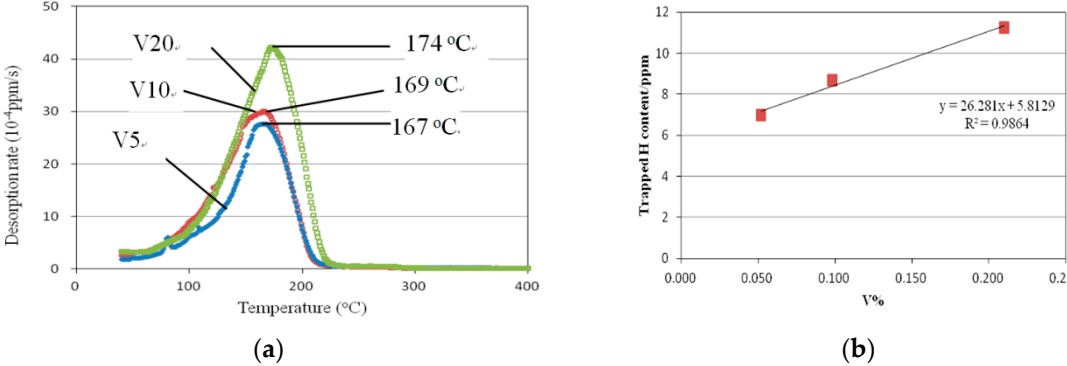

**Figure 5.** (**a**) Thermal hydrogen desorption rate curves of vanadium-added steels, (**b**) Total desorption hydrogen content vs. vanadium content.

## 4. Discussion

### 4.1. Nature of Hydrogen Trapping

The behaviors of hydrogen diffusion were investigated for steels with and without vanadium additions. Different $C_H^S$ values (seen in Table 4) indicated a large amount of hydrogen atom diffusion into the matrix in V-added steels. As above mentioned, the studied steels have similar base chemistry, and $C_H^S$ value for 0.21%V steel is higher than those of vanadium-free and lower vanadium steels. Besides, the amount of trapped hydrogen content increases with an increase in vanadium addition (seen in Figure 5b). It is well known that TiC particles can act as reversible or irreversible traps dependent of its coherence to matrix, nevertheless, they present a strong effect in hydrogen trapping [1,2,8]. In this work, the studied steels have similar Ti contents. All four steels are assumed to have the same level of TiN/TiC with respect to volume fraction and size distributions. It is reasonable that the dominated hydrogen traps in vanadium steels are expected as vanadium precipitates. Since total free N atom had been fixed by Ti and formed as TiN particles due to over-stoichiometric Ti/N ratio in the studied steels, VN cannot be formed. Thus, the vanadium precipitate is assumed in form of VC other than VN particle.

All three vanadium microalloyed steels have nearly identical peak temperatures during hydrogen desorption test, which indicates that vanadium precipitates in these three steels have similar activation energy for detapping of hydrogen which approximates 30 J/mol [12]. Since the peak temperatures of hydrogen desorption rate are rather low, it is reasonable to assume that these vanadium precipitates are reversible hydrogen traps [12]. Lower $D_a$ and higher $H_{total}$ for steel with higher vanadium content indicate that high density of hydrogen trapping sites exists in steel matrix, which is attributed to this clear fact that vanadium can precipitate at tempering temperature of 400 °C, and increasing vanadium addition certainly results in high density of VC precipitates. TEM observations (seen in Figures 6–8) on three vanadium-containing steels validate that the higher volume fraction of VC precipitates presents in steel with higher vanadium content.

From TEM observation results the VC precipitates size distributions for three vanadium steels have been obtained and are shown in Figure 9. The average sizes of VC particles for three different steels are 7.5 nm, 7.3 nm, and 8.4 nm, respectively for 0.052%V, 0.098%V, and 0.21%V steels. The mean size of VC precipitates in three steels do not present obvious difference, but their distributions are clearly different. For 0.052%V steel the VC particles sizes shows uniform distribution. As the vanadium content increases, the distribution of particle size shifts to the smaller size side. Compared with the size distribution of 0.098%V steel, the 0.21% V steel has a larger amount of 'larger size' VC precipitates. When thinking about the effective precipitate size for strengthening mechanism, it shall be at a certain level which thus has the ability of acting as precipitation strengthening. Here an effective size can also be introduced for hydrogen trapping sites. For those precipitates with size either less than or greater than the effective size, they have weaker or even lose the ability of acting as hydrogen

trapping site. In this work, the effective precipitate size can be deduced around 10nm order. Obviously, 0.21%V steel among three investigated steels has the highest frequency of ~10nm precipitates size distribution, in combination with high volume fraction of VC precipitates, showing higher potential of hydrogen trapping.

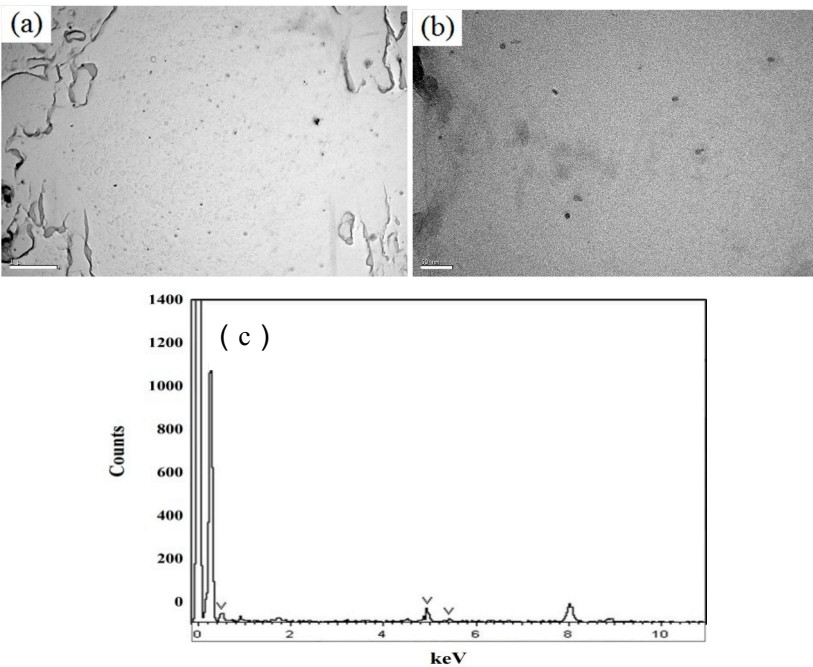

**Figure 6.** TEM observation of 0.052%V TAM steel: (**a**) Microstructure, (**b**) Precipitate, (**c**) Energy Dispersive Spectrometer (EDS) result indicating V precipitates.

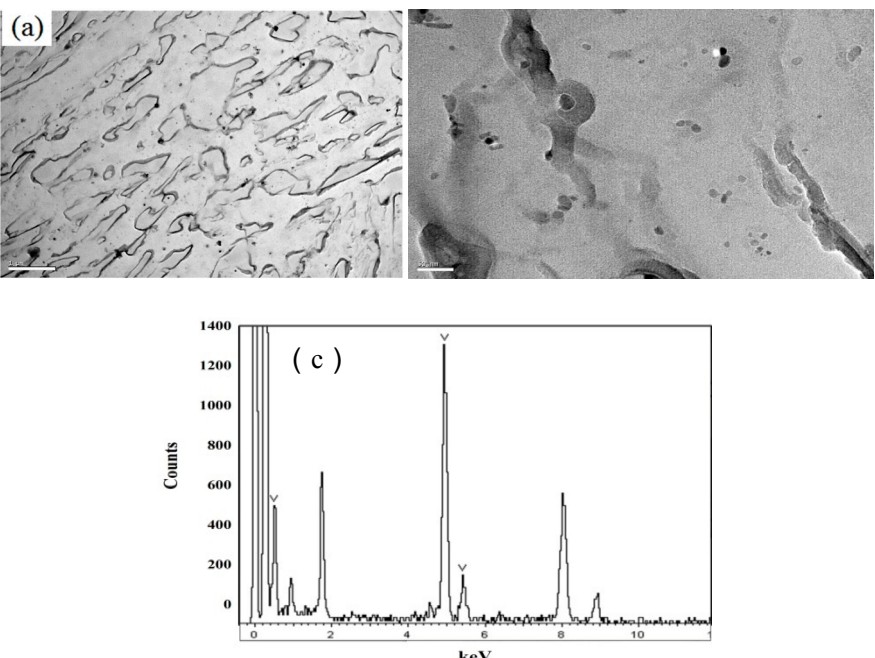

**Figure 7.** TEM observation of 0.098%V TAM steel: (**a**) Microstructure, (**b**) Precipitate, (**c**) EDS result indicating V precipitates.

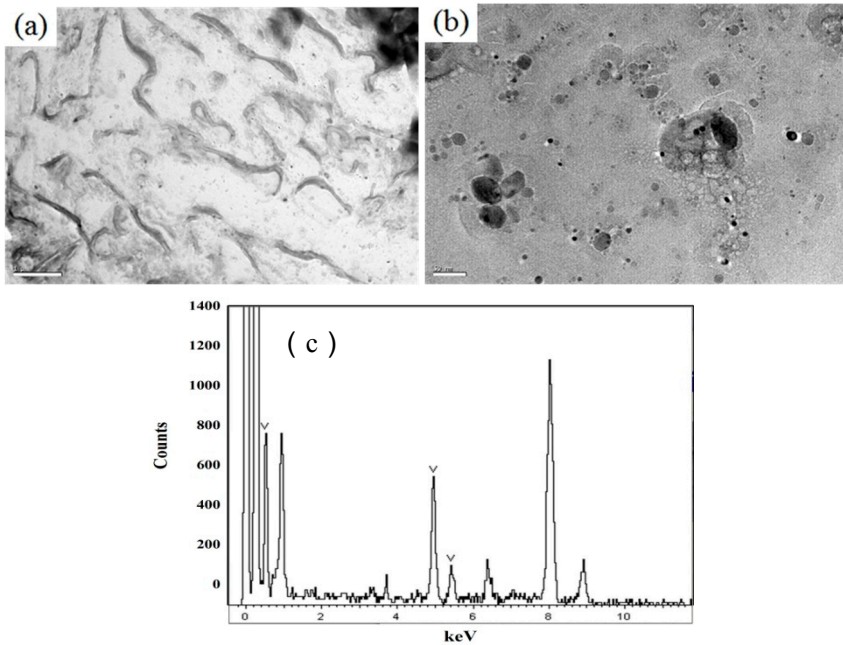

**Figure 8.** TEM observation of 0.21%V TAM steel: (**a**) Microstructure, (**b**) Precipitate, (**c**) EDS result indicating V precipitates.

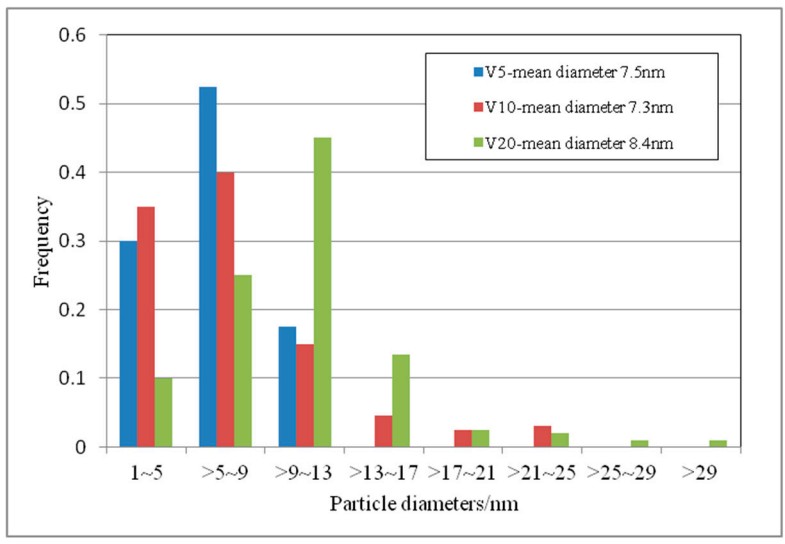

**Figure 9.** Precipitates size distribution of TAM steel with different vanadium additions.

It is known that retained austenite is a kind of strong hydrogen traps which can absorb a large amount of hydrogen [17,18]. The nature of retained austenite determines its ability of trapping hydrogen. Compared with bulk retained austenite as an irreversible trap whose peak temperature of thermal hydrogen desorption approximates 660 °C, thin lath retained austenite presents complete coherence to matrix, and hydrogen atom cannot entry into the retained austenite/matrix interface, thus is considered as a reversible hydrogen trap [18]. In this work, the retained austenite precipitates along the martensite lath during the intercritical annealing process. Therefore, it shall be thin lath retained austenite which is classified into reversible trap.

*4.2. Synergistic Effect of Hydrogen Trap and RA on Ductility Loss*

As mentioned above, when vanadium content is increased from 0.098% to 0.21%, the resistance to ductility loss ratio caused by hydrogen embrittlement effect cannot be improved any longer. Taking the

TRIP effect of retained austenite into account, this phenomenon can be easily explained. When 0.21%V is added in steel, a large amount of carbon solute will be fixed by vanadium and VC precipitates form during 400 °C partitioning(tempering) process, thus the volume fraction of retained austenite is accordingly reduced (seen in Figure 10) in 0.21%V steel. Although high density of VC acting as effective hydrogen trapping sites can improve the resistance to hydrogen induced delayed fracture, the reduced volume fraction of retained austenite decreases TRIP effect. Thus, the ductility accordingly deteriorates compared with other two vanadium steels existing higher volume fraction of retained austenite. Considering the better combination of high strength and good elongation (seen in Table 5) as well as the ability of resistance to hydrogen induced delayed fracture, it is concluded that optimal vanadium addition is 0.098% in this study.

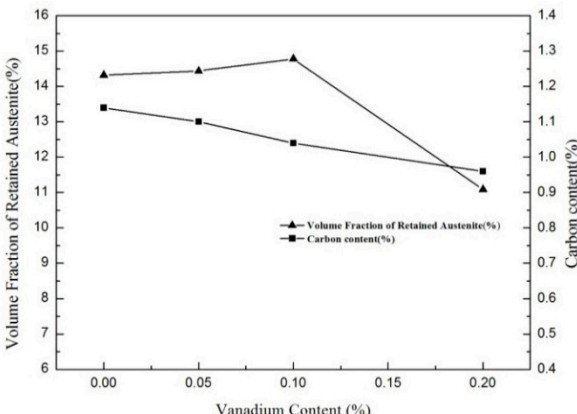

**Figure 10.** Volume fraction of retained austenite and carbon content in retained austenite with different vanadium contents.

As discussed earlier, the hydrogen atom cannot enter into the retained austenite/matrix interface, and then amount of hydrogen absorbed by the lath retained austenite is small, which will leave limited amount of hydrogen into transformed martensite after TRIP effect during tensile process. Thus the retained austenite existing in the studied steels is expected no negative effect on hydrogen embrittlement.

## 5. Conclusions

Hydrogen trapping behavior in TRIP-assisted annealed martensitic (TAM) steels with different vanadium additions had been investigated and nature of hydrogen trapping in these steel are discussed. The important results obtained are as follows.

(1) Hydrogen trap sites of V-added steel with annealled martensite structure are estimated to be vanadium carbide.This vanadium carbide is acting as reversible hydrogen traps. The peak temperature for de-tapping of hydrogen approximates 170 °C for all three vanadium-added steels.

(2) Apparent hydrogen diffusive index $D_a$ and total trapped hydrogen content increasig with vanadium addition. $D_a$ was $1.94 \times 10^{-7}$/cm²·s⁻¹ for 0.21 wt.%V steel, while the value was $8.05 \times 10^{-7}$/cm²·s⁻¹ for V-free steel. The total trapped hydrogen are 7 ppm, 8.6 ppm, and 11.25 ppm for 0.052%V, 0.098%V, and 0.21%V steels, respectively. And there is a clear linear relation between $H_{total}$ and vanadium addition.

(3) High volume fraction of effective-size precipitates are the essential for acting as hydrogen trapping sites.

(4) The hydrogen induced ductility loss $I_D$ was 76.2% for 0.21 wt.%V steel, compared with 86.5% for V-free steel. However, the 0.098%V addition can achieve balancedcombination of high strength and elongation with excellent resistance to hydrogen induced delayed fracture.

**Author Contributions:** Conceptualization, X.Y. and H.Y.; Methodology, X.Y., H.Y.; validation, C.S., L.L. and X.Y.; Formal analysis, X.Y.; Investigation, C.S. and L.L.; Resource, C.S. and L.L.; Data curation, C.S. and L.L.; Writing-original draft preparation, X.Y.; Writing- review and editing, X.Y.; visualization, X.Y.; supervision, H.Y.; project administration, H.Y.; funding acquisition, H.Y.

**Funding:** This research received no external funding.

**Acknowledgments:** The authors would like to thank State Key Laboratory of Vanadium and Titanium Resources Comprehensive Utilization for its financial supports on this work. We also express our appreciation for Prof. Chaofang Dong of USTB for her helpful suggestion on the SSRT.

**Conflicts of Interest:** The authors declare no conflict of interest.

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
