# Peer review of "Hydrogen Trapping Behavior in Vanadium Microalloyed TRIP-Assisted Annealed Martensitic Steel"

_metals, doi:10.3390/met9070741_

Round 1
Reviewer 1 Report
The investigations of hydrogen trapping behavior and hydrogen embrittlement properties of ultra high-strength steels are important to apply for automotive structural parts. Particularly, TRIP-aided steels are expected for the parts. It is supposed that this manuscript will contribute the improvement of hydrogen embrittlement resistance. However, English is poor. The manuscript should be made a native check. The other comments are as follows.
Introduction:
There have been few investigations of hydrogen embrittlement of TAM steel. Authors should add the information of hydrogen embrittlement of TAM steel in introduction.
Lines 79 and 92:
Why were hydrogen charging and SSRT conducted by using different solution?
Line94:
Would you please explain the reason why 500N pre-loading was applied before SSRT?
Line 101-102:
Which part of SSRT test sample were used for TDA samples? The chucking part was not applied plastic strain, whereas a large amount of dislocation was introduced to the deformed part. Therefore, TDA results should be changed and depend on the measuring part of SSRT specimen.
Figure 3 and 4:
Reviewer could not distinguish the lines. Authors should make figures politely.
Line 143:
The microstructure of the steels did not same because the amount and the distribution of VC precipitates should change due to the different amount of vanadium. In addition, retained austenite characteristics were changed because carbon was consumed by the precipitation of VC. Thus, the volume fraction and the carbon concentration of retained austenite should decreased because of VC precipitate.
Discussion:
In the TAM steels, it is necessary to discuss the hydrogen trapping effects of not only VC precipitates but also TiN and retained austenite.
Figure 5:
We could not compare the distribution of particle diameter because the values of horizontal and vertical axes are not same in (a), (b) and (c).
Author Response
point 1: Please carefully check the accuracy of names and affiliations. Changes will not be possible after proofreading.
Response: the names and affiliations of authors are correct. And the telephone number of correspondence is provided in the revised version.

Reviewer 2 Report
1/ Introduction is relatively weak and must be extended and improved. There is nothing on microalloyed (Nb, Ti) TRIP steels. There are a lot of works on this in recent literature (also i Metals, Materials and Applied Sciences). Some additional references are needed. A role of retained austenite and strain-induced martensite in hydrogen trapping should be analysed in more detail.
2/ More experimental details are needed. A final rolling temperature, a final strip thickness, dilatometer conditions, etc ? Without them a study is not reproducible.
3/ The authors refer to SEM and TEM images. Unfortunately, they are not provided in the work. They are obligatory to reflect presented data.
4/ Table 5. Elongation results can not be determined with a such error. What is an error of the determination ?
5/ Axis descriptions should be corrected in all figures for better visibility.
6/ Discussion is limited only to own research results. The references to literaure data are neccessary in a scientific paper.
7/ Discussion and conclusions are focused on VC only. Is it the only reason of the phenomenon. What about the effects of retained austenite and strain-induced martensite ? Cant they be negleted ?
8/ A reference style must be suited and unified to Metals requirements.
Author Response
1/ Introduction is relatively weak and must be extended and improved. There is nothing on microalloyed (Nb, Ti) TRIP steels. There are a lot of works on this in recent literature (also i Metals, Materials and Applied Sciences). Some additional references are needed. A role of retained austenite and strain-induced martensite in hydrogen trapping should be analysed in more detail.
More information had been added to the introduction section. This study focused on the effect of vanadium on hydrogen trapping. Therefore microalloyed (Nb, Ti) TRIP steel are not subject of this study.
2/ More experimental details are needed. A final rolling temperature, a final strip thickness, dilatometer conditions, etc ? Without them a study is not reproducible.
More experimental details had been provided in the revised paper.
3/ The authors refer to SEM and TEM images. Unfortunately, they are not provided in the work. They are obligatory to reflect presented data.
In this work we get the size distribution of VC precipitates from TEM observation . SEM And TEM images are added in the revised manuscript.
4/ Table 5. Elongation results can not be determined with a such error. What is an error of the determination ?
Elongation for specimen without hydrogen charging is great difference from that of specimen with hydrogen charging. Elongation results shown in table 5 are for different vanadium steels with/without hydrogen charging during SSRT.
5/ Axis descriptions should be corrected in all figures for better visibility.
The figure had been modified.
6/ Discussion is limited only to own research results. The references to literaure data are neccessary in a scientific paper.
More discussion had been made.
7/ Discussion and conclusions are focused on VC only. Is it the only reason of the phenomenon. What about the effects of retained austenite and strain-induced martensite ? Cant they be negleted ?
not only VC precipitates but also TiN and retained austenite have hydrogen trapping effects . In this paper TiN effect is not mentioned, since all four steels have similar Ti content and therefore assuming that TiN particles are similar for all steels. Besides, this paper highlights the vanadium effect in hydrogen trapping. The effect of retained austenite is mentioned but not in more detail, since further work should be done to reveal the interaction between reduced volume fraction of retained austenite and increasing volume fraction of VC precipitates.
8/ A reference style must be suited and unified to Metals requirements.
reference style had been modified according to Metals guidance.

Round 2
Reviewer 1 Report
Authors should improve English. Please conduct careful check of English. In the manuscript, both past and present tenses coexist. The reviewer suggest that the manuscript should be checked by native. In addition, authors should carefully make figures. In Figs. 6, 7 and 8, we cannot understand what you want to explain. The other comments are as follows.
Line 112-113:
“in order to stop hydrogen permeation” should be “in order to avoid hydrogen entry”.
Line 114:
Was load of 500 N applied during SSRT? That was a constant load test.
Line 117:
“de-ion” should be “deionized”.
Line174:
Is “all four steels present same J∽” correct? In Table 4, J∽s of TAM-V0, TAM-V5, TAM-V10 and TAM-V20 are 4.36, 4.59, 4.73 and 4.82 x10-10/mol cm-2 s-1. Those are not same values.
Line 192:
According to Table 5, ID of TAM-0V is 86.5 %. Which is correct value 83.4 % or 86.5 %?
Lines 200 and 206:
“isothermal” means “keep the constant temperature”. “isothermal hydrogen desorption rate” is wrong.
Line229-230
Please add the references of activation energy for detrapping of hydrogen in VC precipitates.
Figures 6, 7 and 8:
Scale bars are needed in all photographs. In all figures (c), what is the horizontal and vertical axes?
Figure 9:
In the previous manuscript, the particle diameter of TAM-V20 were between 2 and 32 nm. However, In Fig. 9, the maximum value of particle diameter was 25 nm. Did you remove the data of particle diameter between 26-32 nm? In addition, the range of particle diameter was divided as 1-5, 5-9, 9-13 nm. If the particle diameter was 5.0 nm, how did you count? (count to 1-5 nm or 5-9 nm?). Moreover, “(a) 0.052 %V; (b) 0.098 %V; (c) 0.21 %V” in caption should be removed.
Discussion
It is necessary to discuss the effect of volume fraction of retained austenite on hydrogen embrittlement of TAM steels. It is known that retained austenite absorb a large amount of hydrogen in comparison with ferrite and martensite.
Author Response
Authors should improve English. Please conduct careful check of English. In the manuscript, both past and present tenses coexist. The reviewer suggest that the manuscript should be checked by native. In addition, authors should carefully make figures. In Figs. 6, 7 and 8, we cannot understand what you want to explain. The other comments are as follows.
Extensive English modification/check had been made by a good English skill person. We try use Figures 6-8 to validate the VC precipitates in the steels.
Line 112-113:
“in order to stop hydrogen permeation” should be “in order to avoid hydrogen entry”.
The correction has been made.
Line 114:
Was load of 500 N applied during SSRT? That was a constant load test.
Yes. It was a constant load test. During SSRT the constant 500N load had been applied to specimen until fracture.
Line 117:
“de-ion” should be “deionized”.
The correction has been made.
Line174:
Is “all four steels present same J∽” correct? In Table 4, J∽s of TAM-V0, TAM-V5, TAM-V10 and TAM-V20 are 4.36, 4.59, 4.73 and 4.82 x10-10/mol cm-2 s-1. Those are not same values.
Sorry for the mistake. The word ‘same’ has changed into ‘identical’. Those values are at similar level.
Line 192:
According to Table 5, ID of TAM-0V is 86.5 %. Which is correct value 83.4 % or 86.5 %?
The correct value shall be 86.5%.
Lines 200 and 206:
“isothermal” means “keep the constant temperature”. “isothermal hydrogen desorption rate” is wrong.
Isothermal shall be thermal. It comes from the ‘thermal desorption analysis’.
Line229-230
Please add the references of activation energy for detrapping of hydrogen in VC precipitates.
Literature [12] measures the activation energy of detrapping of hydrogen in VC precipitates.
Figures 6, 7 and 8:
Scale bars are needed in all photographs. In all figures (c), what is the horizontal and vertical axes?
The horizontal axes is KeV, and vertical axes is the count number. Scale bars and names of horizontal and vertical axes are added.
Figure 9:
In the previous manuscript, the particle diameter of TAM-V20 were between 2 and 32 nm. However, In Fig. 9, the maximum value of particle diameter was 25 nm. Did you remove the data of particle diameter between 26-32 nm? In addition, the range of particle diameter was divided as 1-5, 5-9, 9-13 nm. If the particle diameter was 5.0 nm, how did you count? (count to 1-5 nm or 5-9 nm?). Moreover, “(a) 0.052 %V; (b) 0.098 %V; (c) 0.21 %V” in caption should be removed.
The authors try to use this figure to show the size distribution of VC particles. Since only TAM-V20 has coarser VC particles than 26nm, therefore in the revised manuscript the 26-32nm data had been removed. We realize it is wrong doing. A new modified figure with all size distribution is prepared again.
Discussion
It is necessary to discuss the effect of volume fraction of retained austenite on hydrogen embrittlement of TAM steels. It is known that retained austenite absorb a large amount of hydrogen in comparison with ferrite and martensite.
More discussion on effect of retained austenite on hydrogen embrittlement had bee added in the revised manuscript.

Reviewer 2 Report
The authors addressed my remarks properly. New material is added. The work is substantially improved.
Author Response
English checking had made again.
Round 3
Reviewer 1 Report
SSRT
The SSRT was conducted by strain rate of 1x10-6 /s in this manuscript. However, the load was constant at 500 N. These results suggest that the TAM steels did not occur work hardening. Did the increment of load never occur during tensile test in the TAM steels?
Author Response
Q:The SSRT was conducted by strain rate of 1x10-6 /s in this manuscript. However, the load was constant at 500 N. These results suggest that the TAM steels did not occur work hardening. Did the increment of load never occur during tensile test in the TAM steels?
A: thanks very much for your careful check and comments. When consulting the expert in the SSRT field I find I made a mistake during the first response to your question about the reason of applying pre-load.
Actually the applied 500N is pre-load 500N load,not a constant load. The reason to choose a pre-load before SSRT in this work is to eliminate the any possible gap between the specimen and tester, and make zero-point correction.
accordingly this correction has been made to the revised manuscript.

Round 4
Reviewer 1 Report
The modification of reviewer requests has been completed. I suggest that this manuscript is acceptable for the journal.